# Thromboelastography-Guided Anticoagulation in Critically Ill COVID-19 Patients: Mortality and Bleeding Outcomes

Sean Duenas [1,*], Juliana Derfel [1], Margaret Gorlin [2], Serena Romano [1], Wei Huang [3], Alex Smith [3], Javier Ticona [3], Cristina Sison [2], Martin Lesser [2], Linda Shore-Lesserson [4], Negin Hajizadeh [1] and Janice Wang [1]

[1] Department of Medicine, Division of Pulmonary, Critical Care and Sleep Medicine, Donald and Barbara Zucker School of Medicine at Hofstra/Northwell Health, Hempstead, NY 11549, USA

[2] Department of Biostatistics, Feinstein Institute for Medical Research, Manhasset, NY 11030, USA

[3] Department of Medicine, Donald and Barbara Zucker School of Medicine at Hofstra/Northwell Health, Hempstead, NY 11549, USA

[4] Department of Cardiovascular Anesthesiology, Donald and Barbara Zucker School of Medicine at Hofstra/Northwell Health, Hempstead, NY 11549, USA

[*] Correspondence: sduenas@northwell.edu; Tel.: +1-516-465-5400

**Abstract:** Hypercoagulability in COVID-19 patients was associated with increased mortality risk during the pandemic. This retrospective, observational study investigated whether the use of a thromboelastography (TEG)-guided anticoagulation protocol could decrease death and bleeding in critically ill COVID-19 patients. A TEG-guided protocol was instituted in one of two intensive care units. Primary outcomes of composite scores were the following: (0) major bleed and death; (1) death without major bleed; (2) major bleed without death; and (3) no bleed or death. Out of 134 patients, 67 in the TEG group were propensity matched to 67 in the comparator group based on age, gender, body mass index, presence of chronic kidney disease, cardiovascular disease, diabetes, and duration of non-invasive ventilation. There were no significant differences in rates of composite outcomes of bleeding or death in patients managed with or without a TEG-guided protocol ($p = 0.22$, Bowker symmetry testing). Out of the 67 patients in the TEG group, the TEG protocol led to anticoagulation change in 26 patients. Death was lower in this TEG-changed group (54%) compared to the comparator group (81%), although not significant ($p = 0.07$). TEG-guided protocol use did not reduce composite outcomes of death and bleeding, Future studies may further elucidate potential benefits.

**Keywords:** anticoagulants; bleeding; COVID-19; critical care; respiratory distress syndrome; thromboelastography; thrombosis





## 1. Introduction

Coronavirus disease (COVID-19) is associated with a hyperinflammatory response, endothelial dysfunction, and a prothrombotic state that can often progress to acute respiratory distress syndrome (ARDS) and multi-organ failure [1]. Autopsy reports show diffuse alveolar damage and multiple microthrombi in lung vasculature, suggesting pulmonary microthrombosis as a contributing factor to hypoxemic respiratory failure [2]. Hospitalized patients with COVID-19 displayed exceedingly high levels of inflammatory laboratory markers as seen in C-reactive protein, ferritin, and in particular, D-dimer [3]. With the high incidence of thromboembolic events (TEs) observed worldwide in this patient population, an urgent need arose to re-evaluate anticoagulation dosing and monitoring strategies to mitigate risk for TEs in patients with COVID-19.

Since the pandemic, our knowledge of anticoagulation strategies has advanced and guidelines have been published [4,5]. Variable outcomes from a multitude of investigational trials and meta-analyses have demonstrated overlapping conclusions regarding the efficacy and safety of therapeutic anticoagulation used as thomboprophylaxis [6–10]. In the multiplatform REMAP-CAP, ACTIV-4a, ATTACC, and RAPID trials, non-critically ill

hospitalized patients with COVID-19 treated with therapeutic-dose heparin demonstrated increased survival to hospital discharge, along with reduced use of cardiovascular and respiratory organ support compared to those who received thromboprophylaxis. However, this was not demonstrated in the critically ill patients with COVID-19. Rather, therapeutic anticoagulation with heparin in critically ill patients with COVID-19 was associated with a higher risk for major bleeding compared to those treated with thromboprophylaxis dosing [9–13]. The HEP-COVID trial demonstrated that therapeutic-dose heparin reduced the risk of venous TE, arterial emboli, and death in non-intensive care unit (ICU) patients with D-dimer level > 4× the upper limit of normal, but this was not seen in ICU patients [14]. Current guidelines recommend that critically ill patients with COVID-19 receive standard-dose thromboprophylaxis over therapeutic- or intermediate-dose anticoagulation [4,5]. The lack of major benefits of therapeutic anticoagulation in the critically ill COVID-19 patient population is unfortunate. However, a more targeted approach to anticoagulation in the critically ill population could identify patients who may benefit from therapeutic anticoagulation while minimizing their risk of major bleeding complications. Non-specific inflammatory markers, such as D-dimer, C-reactive protein, and fibrin degradation products are not specific enough to reveal the level of COVID-19 coagulopathy [11]. Hypercoagulable states may also not be reflected in activated partial thromboplastic time (aPTT) or anti-factor Xa monitoring strategies. Proposals for the application of thromboelastography (TEG) in monitoring coagulation status in COVID-19 patients have emerged as a result of supportive studies highlighting common trends in TEG monitoring [12,13,15].

TEG is a whole-blood viscoelastic assay that measures coagulation factor function, platelet function, fibrinogen function, total clot strength, and fibrinolysis in a dynamic assessment. TEG may be useful in guiding anticoagulation management and may serve as a guide to improving stratification of hypercoagulable patients who may benefit from therapeutic anticoagulation as thromboprophylaxis. Hranjec et al. demonstrated that care for hospitalized COVID-19 patients utilizing a TEG-guided algorithm had a significantly reduced risk of mechanical ventilation, ICU admission, acute kidney injury, and death compared to patients managed without the TEG-based algorithm [12]. Authors within our group have implemented a TEG-guided anticoagulation protocol in COVID-19 ICU patients, previously published as a case series [13]. Herein, we test the hypothesis that a TEG-guided anticoagulation protocol would reduce the composite outcome of mortality and major bleeding and reduce TEs in critically ill COVID-19 patients.

## 2. Materials and Methods

This was a retrospective study designed to compare outcomes in critically ill COVID-19 patients who had a TEG-guided COVID-19 anticoagulation protocol (Figure 1) vs. routine medical ICU management of anticoagulation [13]. Inclusion criteria were patients 18 years of age and older with COVID-19 requiring an ICU level of care (referred to as critically ill), managed in the ICU for COVID-19 between 1 February 2020 and 1 February 2022, within two large New York metropolitan hospitals of Northwell Health (Long Island Jewish Medical Center and North Shore University Hospital). Exclusion criteria were the following: (1) absence of ICU level of care; (2) ICU admission for a primary reason unrelated to COVID-19 even though the patient was COVID-19-positive (for example, a patient found to be positive for COVID-19 but admitted to the ICU for non-COVID-19-related reasons such as post-operative care following surgery); (3) ICU care requiring the use of extracorporeal membrane oxygenation (ECMO); (4) hospital transfers during their ICU stay; and (5) enrollment into a clinical trial. ECMO patients were excluded due to continuous need for therapeutic anticoagulation. Patients transferred to another hospital were excluded due to loss of electronic medical record (EMR) access and avoidance of differences in clinical practices. Inclusion of three subjects from a previously published case series may have occurred if they met study criteria as they were hospitalized during the same study period.

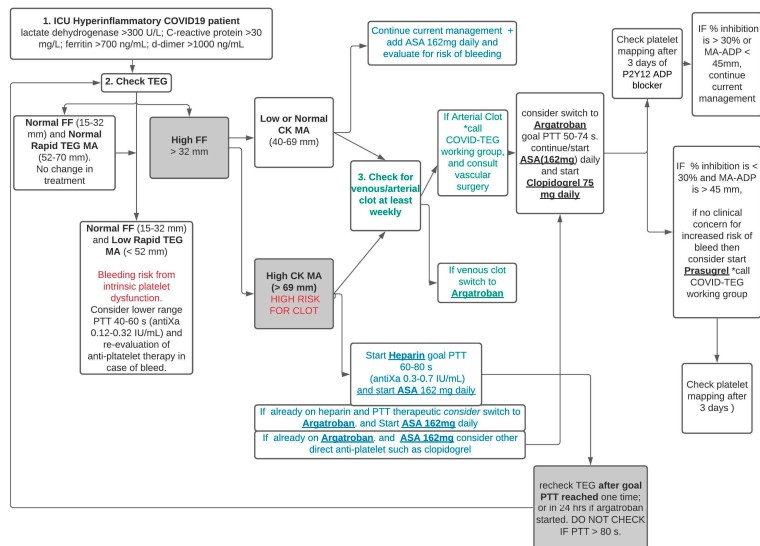

**Figure 1.** Flow chart demonstrating the TEG-guided anticoagulation algorithm. * indicates McNemar's test.

The primary outcome was the composite of death and bleeding. Patients were grouped by whether they were in an ICU with an instituted TEG protocol which was implemented in November 2020, or a comparator arm of ICU patients not managed with a TEG protocol. The comparator group included patients in the same ICU treated before the implementation of the TEG protocol and patients in a similar ICU during the same time period who did not have TEG technology. TEG was measured using the TEG 6S device. A global hemostasis cartridge was used for all patients. As the algorithm did not include monitoring for fibrinolysis, a lysis cartridge, when used, was used identically to the global hemostasis cartridge. Anti-platelet medication assessment was performed using a platelet-mapping cartridge. Both hospitals in the study were tertiary care centers within the same health system, under the same critical care department, and thus, clinical practice and medical management including decisions related to anticoagulation were comparable in both centers.

Our institution's standard of care for hospitalized COVID-19 patients included the use of thromboprophylactic dosing with unfractionated heparin (UFH) or low-molecular-weight heparin (LMWH) based on creatinine clearance (CrCl) and body mass index (BMI). For patients with a creatinine clearance (CrCl) $\leq$ 15 mL/min, UFH dosing was 7500 units subcutaneous every 8 h or 5000 units every 8 h if BMI > 30 kg/m$^2$. Thromboprophylactic dosing was used for patients with a creatinine clearance (CrCl) $\geq$ 15 mL/min, and enoxaparin dosing was 40 mg subcutaneous twice daily if BMI > 30 kg/m$^2$ or 40 mg once daily if BMI < 30 kg/m$^2$. Therapeutic dosing of anticoagulation primarily included treatment with UFH, LMWH, and argatroban. Treatment dosing of UFH was administered using a heparin nomogram protocol. Initial dosing and rate were determined by patient weight with subsequent titrations performed based on aPTT levels. Initial dosing of argatroban was determined by our institutional nomogram which accounted for renal function, hepatic and cardiac comorbidities, as well as critically ill status (greater than two organ dysfunctions). Argatroban titration was based on aPTT levels. Therapeutic enoxaparin was administered every 12 h at a dosage of 1 mg/kg of total body weight. Therapeutic enoxaparin dosing was reduced to daily in those with CrCl < 30 mL/min. During the early pandemic in 2020, universal understanding of COVID-19 was undeveloped; however, the high occurrence of TEs quickly became apparent. Our interdisciplinary team, consisting of critical care intensivists and a cardiovascular anesthesiologist, developed a TEG-guided anticoagulation protocol used alongside the standard of care in the management of critically ill COVID-19 patients.

Patients in the TEG arm were propensity score-matched to subjects in the comparator arm based on variables chosen for risk of COVID-19 severity. These included the following: age, gender, BMI, number of days of non-invasive ventilation (NIV) classified as $\leq$3 days

or ≥4 days, and the presence of pre-existing cardiovascular disease, diabetes, and chronic kidney disease III to V, including dialysis dependency. In consideration of the real-world data captured from our institution's EMR, we chose mortality-related risk factors that were likely to be more prevalent and more widely available to avoid missing data. Increased age, male gender, obesity, and the forementioned comorbidities are associated with poorer outcomes in COVID-19; NIV duration was chosen to reflect the phenotype of the COVID-19 illness [16,17].

Approval was obtained from the Institutional Review Board and the need for informed consent was waived. Research was conducted in accordance with the Declaration of Helsinki. Data were gathered from the EMR and entered in a secure Research Electronic Data Capture (REDCap) database.

The primary outcome was a composite score consisting of death and bleed status by level of severity: 0 = major bleed and death; 1 = death without major bleed; 2 = major bleed without death; and 3 = neither bleed nor death. Individual outcomes of major bleeding, death, and thrombosis were also assessed. Death throughout any time point between ICU admission and hospital discharge were included. Major bleed was defined using the International Society on Thrombosis and Haemostasis (ISTH) criteria as having at least one of the following: (1) fatal bleeding; (2) symptomatic bleeding in a critical area or organ; and/or (3) bleeding causing a fall in hemoglobin level of 2 g/dL or more or leading to transfusion of two or more units of whole blood or red cells [18].

Secondary outcome was TE during hospitalization. TE was defined as venous or arterial thrombosis as evidenced by diagnostic testing. Such testing may have included venous or arterial ultrasound, computed tomography (CT) pulmonary angiogram, other CT scans to evaluate for stroke or other origins of thrombosis, and cardiac catheterization to evaluate for cardiac thrombosis. The TEG protocol recommended evaluation for venous and/or arterial clot if the TEG profile resulted in a high functional fibrinogen > 32 mm. Functional fibrinogen is a proprietary term used by the instrument manufacturer that describes the contribution of fibrinogen to clot strength. Other testing measures mentioned, if performed, were determined by a clinical suspicion or indication according to the clinician.

Propensity score matching (PSM) was used to achieve balance between the TEG and comparator groups on the forementioned variables. The standardized mean difference (SMD) was used to quantify the degree of comparability between the TEG and non-TEG groups. The groups were considered comparable for a given covariate if the corresponding absolute SMD was less than 0.1 [19]. Bowker's test of symmetry was used as the primary statistical test as it tests the equality of the primary outcome score distributions for the paired samples. The Wilcoxon signed-rank test was used to compare the outcome scores via ranks. For TE, McNemar's test for paired binary data was used to assess if the risk of new TE differed in the TEG and comparator groups after matching. A *p*-value < 0.05 was considered statistically significant. All analyses were conducted using SAS version 9.4 (SAS Institute Inc., Cary, NC, USA).

### 3. Results

A total of 556 patients with COVID-19 were identified in the EMR by COVID-19 PCR positivity and lCU location in the two hospitals (Figure 2). Of the 556 patients, 220 patients were excluded; of the remaining 336 patients, 84 patients received TEG analysis during their ICU care, of which 17 were excluded, leaving 67 TEG patients. For the comparator group, 252 patients met the inclusion criteria from which 67 patients were propensity-matched to the 67 TEG patients. The resulting SMD following matching demonstrated that an appropriate level of comparability was achieved for all matched variables (Table 1). Demographics and patient characteristics are shown in Table 2.

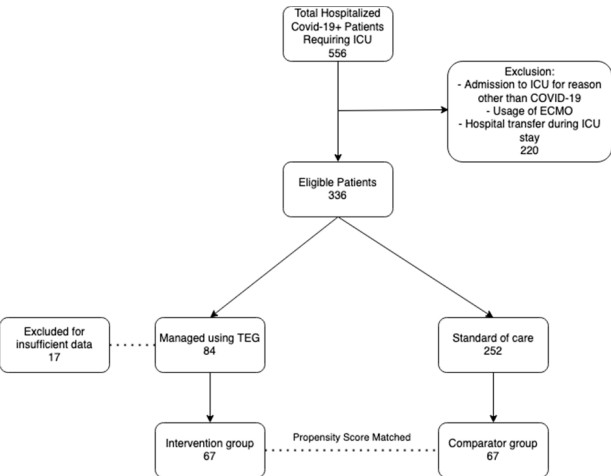

**Figure 2.** Propensity score matching between intervention and comparator groups after exclusion criteria.

**Table 1.** Comparability of the variables used for propensity score matching (PSM) before and after PSM. NIV = non-invasive ventilation.

| Variable | | Standardized Mean Difference |
|---|---|---|
| Age | Before PSM | −0.0340 |
| | Matched | −0.0856 |
| BMI (kg/m$^2$) | Before PSM | 0.0729 |
| | Matched | 0.0754 |
| Gender | Before PSM | −0.0584 |
| | Matched | −0.0312 |
| Diabetes | Before PSM | −0.1845 |
| | Matched | 0.0000 |
| CV disease | Before PSM | 0.0588 |
| | Matched | −0.0336 |
| CKD | Before PSM | 0.0612 |
| | Matched | 0.0501 |
| NIV | Before PSM | 0.0223 |
| | Matched | −0.0682 |

**Table 2.** Demographic, co-morbidity, laboratory (shown as median, lower, and upper quartile range), and COVID treatment data. NIV = non-invasive ventilation. [a] = variables used for propensity score matching between TEG and comparator groups. * = contained missing data.

| | Comparator Group Mean (SD) or No. (%) *n* = 67 | TEG Group Mean (SD) or No. (%) *n* = 67 |
|---|---|---|
| Age, mean (SD) years [a] | 68.5 (11) | 67.4 (11) |
| Sex, No. (%) | | |
| Male [a] | 43 (64%) | 44 (65%) |
| Female [a] | 24 (35%) | 23 (34%) |

**Table 2.** *Cont.*

| | Comparator Group Mean (SD) or No. (%) n = 67 | TEG Group Mean (SD) or No. (%) n = 67 |
|---|---|---|
| Race and ethnicity, No. (%) | | |
| American Indian or Alaska Native | 1 (1%) | 4 (6%) |
| Asian | 2 (3%) | 1 (1%) |
| East Asian | 2 (3%) | 4 (6%) |
| South-Asian | 19 (28%) | 4 (6%) |
| Black or African American | 11 (16%) | 14 (20%) |
| White | 13 (19%) | 24 (35%) |
| Multiracial/Unknown | 19 (28%) | 16 (23%) |
| Hispanic * | 10 (19%) | 10 (15%) |
| BMI [a] | 30.1 (7.6) | 30.6 (8.4) |
| Duration of NIV use, No. (%) | | |
| NIV ≤ 3 days [a] | 52 (77%) | 50 (74%) |
| NIV ≥ 4 days [a] | 15 (22%) | 17 (25%) |
| Co-morbidities, No. (%) | | |
| Cardiovascular disease [a] | 47 (70%) | 48 (71%) |
| Diabetes [a] | 33 (49%) | 33 (49%) |
| Respiratory Illness | 14 (20%) | 12 (17%) |
| Cancer | 4 (6%) | 8 (11%) |
| HIV | 1 (1%) | 1 (1%) |
| Solid organ transplant | 1 (1%) | 2 (3%) |
| Chronic Kidney Disease III—V/Dialysis [a] | 7 (10%) | 6 (9%) |
| Liver disease | 1 (1%) | 3 (4%) |
| Cardiac Arrhythmia | 5 (7%) | 4 (6%) |
| Cerebrovascular disease | 5 (7%) | 3 (4%) |
| History of venous thrombotic event | 2 (3%) | 2 (3%) |
| Laboratory data on ICU admission | | |
| Lactate dehydrogenase (U/L) * | 627 (499–808) | 571 (410–859) |
| C-reactive protein (mg/L) * | 67 (20–158) | 18 (9–80) |
| Ferritin (ng/mL) * | 1068 (538–1741) | 1147 (626–2392) |
| D-dimer (ng/mL) * | 2485 (700–5492) | 2614 (663–5001) |
| COVID-related treatments (not mutually exclusive) | | |
| Remdesevir | 63 (94%) | 59 (88%) |
| Corticosteroids[a] | 66 (98%) | 67 (100%) |
| Tocilizumab | 5 (7%) | 10 (14%) |
| Convalescent Plasma | 0 (0%) | 2 (3%) |
| Monoclonal Antibody | 0 (0%) | 8 (11%) |
| Aspirin | 21 (31%) | 36 (53%) |
| P2Y12 Inhibitor (e.g., clopidogrel) | 3 (4%) | 7 (10%) |
| Exposure to Therapeutic Anticoagulation | 51 (76%) | 47 (70%) |

All patients received either standard prophylactic or therapeutic anticoagulation during their ICU stay. Most patients had received therapeutic dosing of anticoagulation at some point during their hospitalization in both the TEG and comparator groups (70.1% and 76.1%, respectively). Therapeutic anticoagulation may have been initiated based on clinical judgement for concern of TE; this encompasses institutional dosing, as mentioned previously. Other COVID-19-related treatments prescribed included remdesivir and corticosteroids (Table 2).

Composite outcomes, individual outcomes, and treatments received in the group of patients who died and/or had major bleeding are shown in Table 3. Use of TEG-guided anticoagulation protocol did not improve composite outcome of death and major bleed in the TEG group compared to the comparator group ($p$ = 0.22, Bowker symmetry testing). Furthermore, no significant difference in individual outcomes of major bleeding, death, or thrombosis were observed between the TEG-guided anticoagulation group and the comparator group (McNemar's test mid-$p$-values of 0.12, 0.47, and 0.08, respectively). The sample proportion of new TEs was higher in the TEG group (51%) than in the comparator group (34%), although not significant.

**Table 3.** Composite outcomes from 0 to 3 representing severity of outcomes related to major bleed and death analyzed in the TEG vs. comparator groups. Unique outcome of major bleed, death, or thrombotic event analyzed between TEG and comparator groups. [a] = Bowker's test of symmetry, $p$-value = 0.22; * indicates McNemar's test, $p$ = mid-$p$-value.

| | Matched Comparator Group No. (%) $n$ = 67 | TEG Group No. (%) $n$ = 67 | $p$-Value * |
|---|---|---|---|
| Composite outcomes [a] | | | |
| 0—Major bleed and death | 5 (7%) | 9 (13%) | |
| 1—Death without major bleed | 47 (70%) | 39 (58%) | |
| 2—Major bleed without death | 1 (1%) | 3 (4%) | |
| 3—No major bleed or death | 14 (20%) | 16 (23%) | |
| Major Bleed | 6 (9%) | 12 (18%) | 0.12 |
| Death | 52 (78%) | 48 (72%) | 0.47 |
| Thrombotic event | 23 (34%) | 34 (51%) | 0.08 |

Out of 67 patients in the TEG group, TEG-guided protocol led to a change in anticoagulation in 26 patients (we will refer to this group as the TEG-changed group, a subset within all TEG group patients) as the following: therapeutic anticoagulation was initiated with UFH (46%), argatroban (19%), and enoxaparin (8%); UFH was discontinued in five patients (20%) of whom two were switched to either enoxaparin or argatroban; and aspirin was initiated in seven patients. Although enoxaparin was not part of the TEG protocol, management was at the discretion of the clinician. Of the 26 patients in the TEG-changed group, 14 died (54%), 4 (15%) had major bleed, and 12 (46%) had a new TE. In the comparator group matched to the TEG-changed group, 21 died (81%), 3 (11%) had major bleed, and 10 (38%) had a new TE (Table 4). Death was lower in the TEG-changed group compared to the matched comparator group (54% vs. 81%, respectively), however not statistically significant ($p$ = 0.07). There were also no differences in major bleeding or TE.

We acknowledge the limitation of a small sample size which was reduced further following patients who had undergone a TEG-triggered anticoagulation change; however, it revealed that in comparisons of the TEG-changed group vs. TEG group, the number of deaths (54% vs. 72%, respectively) and major bleed (15% vs. 18%, respectively) were lower in the TEG-changed group. TEs were lower in the TEG-changed group compared to the TEG group (46% vs. 51%, respectively).

**Table 4.** Unique outcome of major bleed, death, or thrombotic event analyzed separately between TEG anticoagulation change and comparator groups. * indicates McNemar's test, *p* = mid-*p*-value.

| | Matched Comparator Group No. (%) *n* = 26 | TEG— Anticoagulation Change *n* = 26 | *p*-Value * |
|---|---|---|---|
| Major Bleed | 3 (11%) | 4 (15) | 0.68 |
| Death | 21 (81%) | 14 (54%) | 0.07 |
| Thrombotic event | 10 (38%) | 12 (46%) | 0.66 |

## 4. Discussion

The results of our study demonstrated no significant difference in rates of the composite outcomes of bleeding or death in those managed with or without a TEG-guided anticoagulation protocol. When analyzing death in the TEG-changed group in whom anticoagulation changes were made as a result of the TEG algorithm, there appeared to be overall reduction in death between the TEG-changed group and the matched comparator group, although not statistically significant.

Prior research has demonstrated a consistent relationship between COVID-19 and the increased incidence of TE development, especially in those who progress to critical illness [1–23]. Despite this known increase in the risk of TE, the role of therapeutic anticoagulation in those with severe disease without a diagnosed TE still remains unclear. TEG was performed on patients with hyperinflammatory markers to further select out patients with hypercoagulable states. In patients with COVID-19, fibrinogen levels, while elevated, were not diagnostic of prothrombotic states. The use of MA, which reflects fibrinogen and platelet function, may provide greater insight into the dynamic physiologic state rather than static laboratory tests such as platelet counts and fibrinogen levels. In addition, TEG is more sensitive to heparin compared to standard aPTT measurements and may allow earlier detection of response to anticoagulation [24,25]. While use of a TEG protocol did not demonstrate any difference in primary endpoint, it served to further stratify patients to be considered for escalation of anticoagulation from standard thromboprophylactic dose and to trigger TE workup if not already performed. High functional fibrinogen served as the branch point that differentiated patients with overall hyperthrombotic clot strength vs. those with normal clot strength reasoned for escalation of anticoagulant therapy for those potentially at greater risk. Subsequent anticoagulant monitoring plus monitoring anti-platelet medication using platelet mapping, if added, was another feature of the TEG-guided algorithm to consider.

The practice of anticoagulation escalation was performed with supportive TEG profiles and clinical expertise at a time when there was limited knowledge and paucity of clinical trials in this field. We recognize now that many trials have demonstrated a lack of efficacy for therapeutic anticoagulation in critically ill patients; however, the purpose of examining the use of TEG in the critically ill COVID-19 patient population was to determine if integrating its use into clinical care would have a different impact by selecting out patients with TEG evidence for hypercoagulability.

We acknowledge several limitations in this real-world study, including the small sample size and retrospective nature of the study. Thrombosis screening was also performed when clinically indicated in both arms and additionally when the algorithm called for it in the TEG group. The added surveillance in the TEG protocol may have resulted in the higher rates of subclinical TE. In the comparator arm, 76% of patients were also exposed to therapeutic anticoagulation during their ICU care, which reduced the generalizability of the study since therapeutic anticoagulation occurred just as frequently as in the TEG group (70%). Duration of anticoagulation was also extremely variable amongst patients.

Death was reported for the entire hospitalization and varied greatly, which may have influenced results as well; hospital length of stay until time of death ranged from 5 to 105 days (median of 19 days) in the TEG group compared to 2 to 86 days (median of 22 days).

Proposed mechanisms for COVID-19-induced thromboembolism suggest pathogenesis through early immune dysregulation and endothelial dysfunction, which can result in the expression of prothrombotic molecules and platelet activation with resulting platelet aggregation and thrombosis [26]. Because our protocol relied on serologies drawn after patients had likely already reached clinical deterioration and critical illness, one could speculate that we surpassed the window of benefit with TEG-guided anticoagulation.

While a standard TEG anticoagulation algorithm was provided, clinical decision making may have superseded TEG protocol-related recommendations. Additionally, due to the study spanning over 2 years since the start of the pandemic, there may have been temporally related differences in COVID-19 treatments, such as the implementation of vaccination, monoclonal antibodies, convalescent plasma, and immunomodulators, as well as evolving COVID-19 variants (progression from alpha, beta, and gamma strains to delta and omicron). While the patients receiving certain COVID-19 therapies were small in number, we recognize that cumulative therapies in the TEG group did exceed those observed in the comparator group. These treatment variables may have limited propensity matching; however, we chose the most commonly occurring variables that were already known to influence mortality at the time of the study while not exceeding the statistically recommended number of variables.

### 5. Conclusions

Our study showed insufficient data to demonstrate the benefit of a TEG-guided anticoagulation algorithm in improving composite or individual outcomes of death, major bleeding, or thrombosis in critically ill COVID-19 patients admitted to the ICU. The results of our study, in addition to data from prior anticoagulation studies in non-critically ill COVID-19 patients, highlight the importance of patient selection and timing of anticoagulation prior to a critical point in the thrombosis and hypercoagulability cascade. Future studies are needed to determine the efficacy of TEG-guided management in reducing death and TE while minimizing major bleeding in critically ill COVID-19 patients.

**Author Contributions:** Data collection, S.D., J.D., S.R., W.H., A.S., J.T., N.H., and J.W.; methodology, L.S.-L., N.H., and J.W.; data analysis, M.G., C.S., M.L., N.H., and J.W.; writing—original draft preparation, S.D., J.D., M.G., L.S.-L., N.H., and J.W.; writing—review and editing, S.D., J.D., L.S.-L., N.H., and J.W. All authors have read and agreed to the published version of the manuscript.

**Funding:** This research received no external funding.

**Institutional Review Board Statement:** This study was conducted in accordance with the Declaration of Helsinki and approved by the Institutional Review Board of Northwell Health Systems (Protocol Code #21-0225, date approved 10 March 2021).

**Informed Consent Statement:** Patient consent was waived due to the retrospective nature of the study.

**Data Availability Statement:** The data are not publicly available due to institutional IRB regulations and adherence to protected health information policies.

**Conflicts of Interest:** The authors declare no conflicts of interest.

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
