# Peer review of "Thromboelastography-Guided Anticoagulation in Critically Ill COVID-19 Patients: Mortality and Bleeding Outcomes"

_2673-527X, doi:10.3390/jor4010001_

Round 1
Reviewer 1 Report
Comments and Suggestions for Authors
Content suggestions:
1. How did the data about the previous antithrombotic treatment influence the protocol ?
2. TEG is measured at 37 degrees of Celsius. However, these patients were instable in terms of the increase of temperature. Thus, did the authors standardize their results and if yes, how ?
I sincerely appreciate each new information about the life-threatening infection caused by SARS-CoV-2 virus. Therefore, the manuscript can be published after the response to the content suggestions of the reviewers.
Reviewer 2 Report
Comments and Suggestions for Authors
A TEG-guided protocol was instituted in one of two adults intensive care units. Out of 134 patients, 67 in the TEG group were propensity matched to 67 in the comparator group based on different parameters. Bottom line: TEG-guided protocol use did not reduce composite outcomes of death and bleeding.
First of all, what is different form the report ref#13? Overlap of included patients in the two reports MUST be clearly indicated.
The retrospective design is a major weakness as acknowledged L.256, and results in many exclusions.
« …clinical-decision L.272 making may have superseded TEG protocol-related recommendations »: How often could it have occurred?
Actual administered anticoagulation regimens to be thoroughly described.
“All patients received either standard prophylactic or therapeutic anticoagulation L.179 during their ICU stay. Most patients had received therapeutic dosing of anticoagulation 180 at some point during their hospitalization in both the TEG and comparator groups 181 (70.1% and 76.1%, respectively). Therapeutic anticoagulation was initiated based on clin- 182 ical judgement for concern of TE.” A more detailed account of anticoagulation is required. Actual administered anticoagulation regimens are to be thoroughly described.
“The interdisciplinary team L.117 consisting of critical care intensivists and a cardiovascular anesthesiologist developed a L.118 TEG-guided anticoagulation protocol used alongside standard of care in the management 119 of critically ill COVID-19 patients. “
TEG: which device? Which sets of reagents / cartridges (if TEG6S)? Parameters in the figure #1 to be fully explained in the text.
It seems that this was done rapid TEG and sometimes Platelet-Mapping; Parameter: MA only: Please confirm / explain / comment
MA is related to fibrinogen levels and platelet counts: the authors should comment on the option that an algorithm based on those two readily available and robust lab results would do the job at least as well.
« TEG is a whole blood viscoelastic assay that measures coagulation factor function, L.70 platelet function, fibrinogen function, total clot strength, and fibrinolysis in a dynamic L.71 assessment. »: viscoelastometric seems to be more appropriate when talking about the measurements of mechanical properties of a forming and evolving clot; platelet ‘function’ is mainly if not only the mechanical action on fibrin fibres; what is fibrinogen ‘function’?
What is known in the literature about sensitivity of TEG to heparins?
Which were the SARS-CoV-2 strains during the time-period of the study? (L.275)
No standard lab data are provided. D-dimers, aPTT, anti-Xa: which specific assays?
How action #3 in Fig.1 was actually performed?
L.33 thrombotic microangiopathy to be explained
I don’t see how “microthrombi [can] lead L.267 to further activation of coagulopathic pathway » and I don’t know the concept of ‘coagulopathic pathway’
“…exceedingly high levels of hyper- L.37 inflammatory and hypercoagulable laboratory markers as seen in C-reactive protein, fer- L.38 ritin, and in particular, D-dimer”
What is ‘hyperinflammation’ ?
D-dimer is not a biomarker of hypercoagulability.
L.259 cannot start with 76%
What are the differences between ‘race’ and ethnicity? (table 2) By the way ‘race’ is not a scientifically grounded concept. Ancestry would be more appropriate, but for what purpose? (I have failed to find any use in the text)
Introduction and discussion sections can be much shortened. In short, lab assessment of the coagulation phenotype would theoretically be useful to guide anticoagulation. To what extent TEG is a good candidate is highly debatable.
Authors’ addresses are incomplete.
(New York metropolitan hospitals of Northwell Health (Long Island Jewish Medical Center and North Shore University Hospital) )
Round 2
Reviewer 2 Report
Comments and Suggestions for Authors
What is known in the literature about sensitivity of TEG to heparins?
TEG is sensitive to “heparins” and the R time will prolong in the presence of unfractionated heparin and LMWH’s. It is more sensitive to LMWH’s and will detect prolongation, whereas standard aPTT measurements are not as sensitive. (Tekkesin 2016, Zmuda 2000).
The above discussion has been added in L293-298.
> disagree - not supported enough, systematic review of the literature required, at least for TEG 6s device and Global hemostasis cartridge
When the effect of anti-platelet medication was assessed in patients with TE: what does it mean?
The term “fibrinogen function” refers to the combination of fibrinogen “level” and the “function” of fibrinogen, or dysfibrinogenemia.
I strongly suggest not using it: levels are always measured measured in daily clinical practice with a functional assay, that is clotting in presence of added thrombin (Clauss)
